# What Do LLMs Understand About International Trade? Introducing TradeGov Dataset for International Trade Q&A Evaluation

## Abstract

Given the constant flux in the world of geopolitics, staying up to date and compliant with international trade issues is challenging. But exploring if LLMs can aid this task is a frontier hither to unexplored in the LLM evaluation literature - primarily due to the lack of a dataset set for benchmarking the capabilities of LLMs on questions regarding international trade subjects. To address this gap, we introduce TradeGov - a novel, human audited dataset containing 5k international trade related question-answer pairs across 138 countries, created using ChatGPT based on the Country Commercial Guides on the International Trade Administration website. The dataset achieves 98% relevance and faithfulness and doesn't show any systematic biases along macroeconomic and geographical dimensions, lending itself to equal applicably for LLM assessment across countries. Testing the performance of ChatGPT 4o on this dataset - marking the first systematic evaluation of LLMs for answering questions about international trade - we find that it achieves 84% accuracy. However, we also show that ChatGPT 4o has bias, it performs better for countries with greater ease of business, higher GDP and higher trade shares. The TradeGov dataset thus fills a critical gap in the LLM evaluation literature and paves the way for greater understanding of how LLMs can assist in navigating the complex international trade landscape.

## 1 Introduction

In an increasingly globalized world, understanding and complying with international trade matters is crucial for both governments and businesses alike. For governments, it is essential to strike a balance between protecting domestic markets and integrating with the global economy. For businesses, staying abreast with international trade affairs is crucial for a. mitigating and minimizing losses due to fines on business operations and lost business opportunities, while b. also maximizing profits by taking advantage of legal opportunities for cross-border trade. However, navigating the complex legal landscape of international trade requires specialized legal expertise, which is not equitably available to all. Illustratively, larger businesses have the capital to leverage the expertise of lawyers specializing in the trade of a particular country (say India) while a small businesses are unlikely to have similar expertise, thus making them comparatively less competitive in the global economy. Large Language Models (LLMs) have the potential to bridge this gap by offering reliable information regarding international trade. If LLMs can effectively interpret and provide information about international trade, they could assist both small and large businesses in understanding regulatory requirements and expanding into global markets. LLMs could also aid government entities in navigating complex policy negotiations and red tape associated with international trade regulations. Therefore, it is important to evaluate how well LLMs can handle questions related to international trade. However, the current

LLM evaluation literature does not address the capabilities of LLMs for question-answering in this domain. A primary impediment is the lack of a dataset for benchmarking the performance of LLMs on Q&A tasks related to international trade. We address this gap by introducing a novel dataset on international trade called TradeGov - constructed by leveraging Retrieval Augmented Generation (RAG) with ChatGPT 4o to generate international trade question and answer pairs using the Country Commercial Guides on the International Trade Administration website. This paper describes the construction of this dataset and implements a framework for assessing the quality and biases (along global macroeconomic and geographical inequalities) of the Q&A pairs generated. It then carries out a novel LLM benchmarking exercise by evaluating the performance of ChatGPT 4o for answering questions related to international trade.

## 2 Literature Review

This paper situates itself at the intersection of three fields: applying LLMs to law, international trade law, and creating novel datasets for LLM benchmarking. LLMs have been applied to various legal tasks such as summarization [19][10], Q&A [16][2], legal judgment prediction [7], text extraction, and reasoning. Numerous datasets support these tasks, including corpora for argument mining (Demosthenes, CDCP), legal case analysis (CaseHOLD, European Court of Human Rights Dataset), contract review (CUAD, ContractNLI), and regulatory analysis (EUR-Lex-Sum, Caselaw Access Project). However, there is a notable gap in datasets focused on international trade law, which this paper addresses. Relatedly, the use of LLMs in international trade law has limited literature, with most research focusing on AI regulation from a international trade perspective [3] or the impact of generative AI on international trade negotiations [1]. However, to the best of our knowledge no paper systematically addresses the ability of LLMs to answer international trade law related queries - a gap which this paper addresses through the creation of the TradeGov dataset and evaluating ChatGPT-4o on the same.

### 2.1 TradeGov Dataset : Construction Methodology

To construct an open source benchmark dataset for measuring the performance of LLMs on international trade Q&A, four constraints were at the fore for the source data: 1) it must be non-proprietary, 2) it must be from a reliable, legally trusted source, 3) it should allow periodic updates to reflect changes in trade regulations across 150 countries, and 4) it must cover both high and low income countries. Ideally, this would involve extracting relevant information from each country's official government websites. However, this is an extremely difficult task because the degree to which the international trade regulation information is available for a country varies greatly. For instance in South Korea, the Korean Law Information website has all the required information in highly structured and searchable manner, but for Brazil, the information is neither available in a consolidated or well structured / searchable fashion. Thus, we forego this methodology to avoid bias in the quality and amount of information collected for each country due to a country's online government infrastructure. Using international trade books was ruled out due to copyright concerns. Thus, we determined the Country Commercial Guides on the International Trade Administration website maintained by the US government [27] to be the most suitable source. The website contains information on "market conditions, opportunities, regulations, and business customs prepared at the U.S. Embassies worldwide by Commerce Department,State Department and other U.S. agencies"[27] regarding all countries with any trade relation with the US.It is suitable because 1) it is not a proprietary domain and thus can be scraped and used for making a dataset(double checked with lawyers); 2) is considered to be a reliable source with up-to-date information for international trade regulation by lawyers; 3) updates information regularly and 4) it covers 150 countries. This data source also has the added advantage that is covers key World Trade Organization agreements / treaties as well. However, this website offers a trusted and comprehensive but limited high-level overview of the international trade landscape, with drawbacks including: 1) lack of information on the U.S. domestic trade policies, 2) potential omission of trade agreements to which the US is not a party, and 3) it being in English due to which nuances found in local language sources are lost. Despite these limitations, we argue that this provides a valuable starting point for evaluating LLM performance on international trade related questions at scale, given the current gap in the literature regarding the same.

To create the Q&A dataset, we scrape the information from the website for Customs, Regulations and Standards section for 150 countries. For each country, the website contains information about 11

Table 1: TradeGov Evaluation: Q&A Quality and Bias Assessment

| Type | Mean | Correlation | Correlation | Correlation |
|---|---|---|---|---|
| Metric | | Ease of Doing Business | GDP per capita | Trade % of GDP |
| Relevance | 0.976657 (0.15) | 0.089 (0.325) | -0.138 (0.156) | -0.040 (0.690) |
| Question Specificity | 0.698419 (0.45) | 0.374 (0.000) | -0.376 (0.000) | -0.174 (0.083) |
| Answer Specificity | 0.981363 (0.13) | -0.045 (0.621) | 0.046 (0.638) | 0.092 (0.365) |
| Faithfulness | 0.977786 (0.15) | -0.168 (0.062) | 0.076 (0.435) | 0.053 (0.597) |
| Scraped Text Length (characters) | 3520 (4005.01) | -0.350 (0.000) | 0.270 (0.005) | -0.020 (0.830) |
| # Questions per Country | 36 (16.27) | -0.180 (0.045) | 0.140 (0.141) | -0.190 (0.055) |
| # Categories per Country | 7 (2.12) | -0.170 (0.056) | 0.170 (0.087) | -0.150 (0.129) |

Brackets in mean column/s contain standard deviation and for correlation columns contain p-values.

categories : Trade Barriers, Import Tariffs, Import Requirements and Documentation, Labeling and Marking, Export Controls, Temporary Entry, Prohibited and Restricted Items, Customs Regulations, Standards for Trade, Trade Agreements and Licensing Requirements for Professional Services. To create Q&A pairs, we use ChatGPT-4o and follow these steps: 1) We provide ChatGPT-4o with text scraped from each category and country combination; 2) Using an optimized prompt (see Appendix Figure 3), we instruct ChatGPT-4o to generate question-answer pairs based solely on the provided scraped text; to ensure that the generated Q&A pairs come only from the scraped text and not the model's internal world knowledge, we apply Retrieval-Augmented Generation (RAG) principles and ask the ChatGPT to provide exact quotes with citations for each answer it creates. To improve the quality and relevance of the generated Q&A pairs, we used in context learning (ICL) examples along with auto prompt tuning to create a dataset of 5,100 question-answer pairs regarding international trade (see Appendix Table 5 for a sample of generated Q&A pairs in the dataset) ([1]

## 2.2 Dataset Evaluation

Having constructed the data, we determine the quality of the generated Q&A pairs using a human-in-the-loop audit with the following four criteria: 1) **Answer Relevance**: is the answer relevant to the question asked?; 2) **Faithfulness**: is the question-answer pair created only from the scraped text provided? ; 3) **Question Specificity**: is the created question very broad? ; 4) **Answer Specificity**: is the generated answer generic and lacking in details? Our dataset of 5,100 questions achieved 98% Faithfulness , Relevance, and Answer Specificity with 69% specific questions (see Table 1). If a Q&A pair lacks relevance, faithfulness and has a vague answer, it is removed from consideration, leaving us with 4992 Q&A pairs. This dataset consists of approximately 36 questions per country across 7 categories on average (see Table 1). The subject matter of majority of the Q&A pairs is import tariffs, trade standards, trade agreements, import requirements and documentation and trade barriers (see Appendix Table 3 for more details).

## 2.3 Bias Evaluation

Given that our dataset covers 150 countries, there is potential for representation biases. Particularly, it is possible that the dataset has a higher quantity and quality of Q&A pairs for nations that have 1) policies well documented on the internet, 2) are wealthier and 3) have trade as a big part of their economy. For each country in the dataset, we investigate these three potential biases using the correlation between country level average values for the dataset evaluation metrics mentioned in section 2.2 and three macro-economic indicators[2]: 1) **Ease of Doing Business Index**: A proxy for the level of digital documentation of a country's rules and regulations; 2) **GDP per capita (GDP PC)**: An indicator of economic development and 3) **Trade as %age of GDP**.

Referring to Table 1, we see that there is neither any statistically significant correlation between the dataset evaluation metrics and 3 macroeconomic indicators nor is there any discernible geographical bias (see Figure 1 and Appendix Figure 4, 5, 6) in the number of Q&A pairs created for a country. The only exception to this is Question Specificity - which has statistically significant but weak positive correlation with the Ease Of Doing Business Index and weak negative correlation with GDP PC. This

---

[1]Due to a country name mapping error, the dataset currently has coverage for 138 out for 150 countries. These geographies will be included in forthcoming versions of the dataset.

[2]Source : World Bank Open Data (https://data.worldbank.org/)

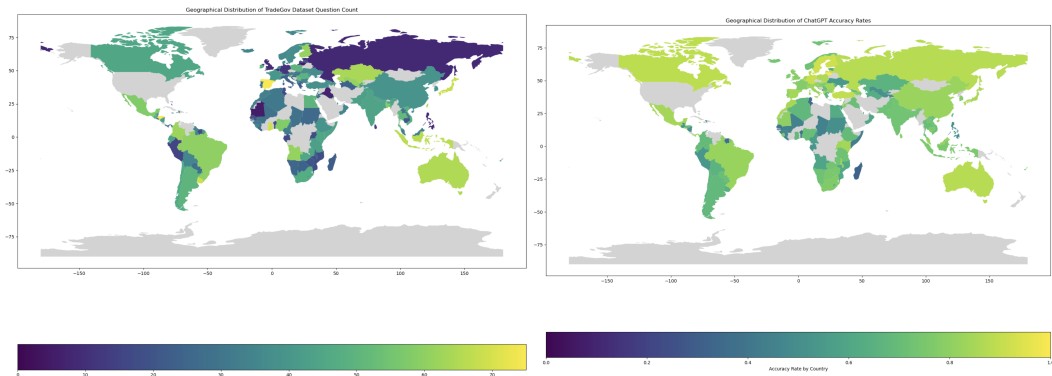

Figure 1: TradeGov Question Count Distribution  Figure 2: ChatGPT Accuracy Rate Distribution

Table 2: ChatGPT Evaluation: Answer Quality and Bias Assessment

| Type | Mean | Correlation | Correlation | Correlation |
|---|---|---|---|---|
| Metric | | Ease of Doing Business | GDP per capita | Trade % of GDP |
| Null Rate | 0.163 (0.369) | 0.51 (0.0) | -0.28 (0.004) | -0.18 (0.07) |
| Accuracy | 0.845 (0.361) | -0.586 (0.0) | 0.345 (0.0) | 0.236 (0.018) |
| Completeness | 0.740 (0.438) | -0.539 (0.0) | 0.377 (0.0) | 0.22 (0.028) |
| Specificity | 0.400 (0.4900) | 0.229 (0.01) | -0.111 (0.255) | -0.218 (0.029) |
| Longest Substring Overlap Length (Memorization proxy) | 14.000 (16) | -0.36 (0.0) | 0.19 (0.04) | 0.1 (0.331) |

Brackets in mean column/s contain standard deviation, and for correlation columns, contain p-values.

128 finding holds true across all information categories (see Appendix Table 4 for details). [3]. Notably,
129 there is a statistically significant (weakly) positive correlation (0.3 at the 0.001 level) between the
130 average length of the website text scraped and the number of Q&A pairs generated for a country. The
131 average length of the text scraped is also statistically significantly: 1) negatively correlated with the
132 Ease Of Doing Business Index and 2) positively correlated with GDP PC (see Table 1 and Appendix
133 Figure 4). However, interestingly, the number of Q&A pairs generated for a country does not display
134 a similar correlation - it is only weakly negatively correlated with the Ease Of Doing Business Index
135 (at the 0.5 level of significance; see Table 1) ; we hypothesize this is due to the construction of our
136 prompt which limits the number of Q& A pairs created for any country and topic to the range 5 to 10.

137 The above results are encouraging as they demonstrate that the TradeGov dataset does not have any
138 obviously discernible biases in it, which helps the dataset have broad and credible applicability across
139 all countries for international trade Q&A related tasks.

## 3   LLM Benchmarking Methodology

141 Equipped withe the TradeGov dataset, we benchmark the performance of ChatGPT-4o for answering
142 international trade policy related questions (see Appendix 1 for the prompt). We evaluate the
143 responses generated across 4 dimensions: 1. **Accuracy**: Does the answer generated by the LLM
144 contain the key facts in the benchmark TradeGov answer? ; 2. **Completeness**: Does the LLM answer
145 contain all the details mentioned in the benchmark TradeGov answer? ; 3. **Specificity**: Does the
146 LLM answer contain unnecessary details? ; 4. **Null response rate**: Is the answer "I don't know"? We
147 use a human-in-the-loop audit to evaluate the LLM generated answer against the answers mentioned
148 in the TradeGov dataset across all four criterion.

## 4   Results

150 Examining Table 2, we see that ChatGPT-4o has a Null Rate of  16%. Examining the relationship
151 between null response rates by country and the macro economic indicators mentioned in section 2.3,
152 we see that there is a strong positive correlation (0.5; statistically significant) between null rate and

---

[3]Note: Topic modeling for each country using Latent Dirichlet Allocation didn't show any discernible
differences in the content of the text scraped across countries and thus is omitted from discussion here.

the Ease of Doing Business Index and 2) a weak but negative correlation between null rate and GPD PC (see Table 2). Thus, we conclude that ChatGPT-4o is more likely to respond with "I don't know" for countries with lower online policy documentation and lower economic development.

After filtering out null responses, we are left with 4200 questions. On this subset, ChatGPT-4o achieves an accuracy of 84%, completeness of 74% and specificity of 40%. To determine if these results are on account of ChatGPT-4o parroting text it has memorized from the International Trade Administration website, we split the answer for each query into half and ask ChatGPT-4o to complete the sentence. Then, we use longest sub-string match and sub-string overlap to determine if it outputs text exactly matching the one found on the International Trade Administration website or not. Table 2 and Appendix Fig. 9 show this to not be the case - for majority of the dataset, the longest sub-string match is less than 20 characters.

Given that ChatGPT-4o is likely to have representation biases of the nature mentioned in section 2.3 - we apply the same bias evaluation frame work to analyze the answers generated by ChatGPT-4o. We compute the per country mean values of Null Rate, Accuracy, Completeness, Specificity and Longest sub-string overlap length and measure the correlation of the same with the Ease of Doing Business Index, GDP PC and Trade share % of GDP. We find that there is statistically significant evidence of ChatGPT-4o performing better for countries with greater ease of business, higher GDP PC and a larger share of trade in their GDP, with worse performing countries being concentrated in Africa (see Figure 2; Table 2; Appendix Figure 8). Particularly, the null rate, accuracy and completeness are statistically significant, strongly negatively correlated with the Ease of Doing Business Index, signaling that the lower the digital documentation for a country, the worse ChatGPT-4o performs. They are also statistically significantly (but weakly) positively correlated with GDP PC, implying higher a country's per capita income, the better ChatGPT-4o performs. Accuracy and completeness are also statistically significantly (but weakly) positively correlated with Trade %age of GDP - indicating that ChatGPT-4o knows more about the trade regulation of countries that trade more. Lastly, answer specificity being weakly positively and negatively correlated with Ease of Doing Business and Trade % of GDP (see Table 2; Appendix Figure 8) respectively potentially indicates that ChatGPT-4o generates answers with more details than needed for countries with lower online documentation and smaller trade shares as it is more uncertain of its knowledge and thus wants to cast a wider net while answering. We leave investigations into these claims to forthcoming versions of the paper.

# 5   Conclusion

We introduced the TradeGov dataset - the first human-audited, open source dataset for evaluating the performance of LLMs within the domain of on international trade related Q&A. Using this, we were able to show that while current state of the art LLMs can achieve high performance ( 84% accuracy) in answering factual questions about international trade, this performance is not equitable and is biased in the favour of countries with greater ease of business, higher GDP and higher trade share. To provide continued support for such analysis, improving the generation of Q&A pairs for the TradeGov dataset iteratively is key. More context needs to be added to the questions to reduce ambiguity and improve Question Specificity. The adherence of the Q&A generation to instructions regarding no duplication needs to be addressed as well - despite asking the model to not generate duplicate question, we get questions which are very similar in meaning (Ex: "What is the role of INMETRO in Brazil's regulatory regime?" ; "What is INMETRO responsible for in Brazil according to international trade law?" are the same question). Furthermore, most questions are factual (96% are "what" questions) and focus on recalling information rather than understanding the international trade landscape. The TradeGov dataset also lacks information regarding agriculture - only 2% of the queries include agriculture or food. This is a critical gap for emerging markets where majority of trade policies deals with agriculture. We shall use few-shot ICL and iterative prompt tuning to improve question specificity, reduce duplication and encourage generation of more cause and effect related questions. To improve the grading of the questions, we will engage lawyers next as opposed the current non-expert auditors. This is especially important because given the low specificity, the subject matter expertise of a lawyer is required to understand if the additional generated facts generated by LLMs - not contained in the TradeGov dataset - are correct or not. This will also aid in establishing a robust human base line for answering international trade related questions, against which the performance of LLMs can be better contextualized.

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

# Appendix

Table 3: Number of Questions per Category

| Category | Num. Questions |
|---|---|
| import-tariffs | 768 |
| standards-trade | 763 |
| trade-agreements | 627 |
| import-requirements-and-documentation | 626 |
| trade-barriers | 606 |
| customs-regulations | 471 |
| temporary-entry | 379 |
| licensing-requirements-professional-services | 329 |
| labelingmarking-requirements | 301 |
| prohibited-restricted-imports | 122 |

## .1 Prompt for evaluating the performance of ChatGPT on TradeGov Dataset

```
Question : What is required for all vehicles, both new and used, that are imported
    into Russia according to technical regulation TR TS 018-2011?

Prompt :

f"""Answer the following question. If you don't know the answer to a particular
    question, answer with 'I dont know'.\nQuestion: {question}\nAnswer:"""
```

Figure 3: Prompt Template for TradeGov Dataset Q&A Generation

```
Example Text Extract: The following labeling information must be in Croatian on the
    original package of products subject to quality control: name of the product;
    full address of the producer or full address of the importer; net quantity,
    weight, or volume; ingredients; usage and storage particulars; and any
    important warnings about the product for the consumer. Technically complicated
    products must include instructions for use, the manufacturers specifications, a
     list of authorized maintenance offices, warranty, and other applicable data.
    Every certified product must carry a CE mark indicating that the product has
    undergone appropriate testing and that it conforms to the provisions of the
    relevant regulations. Foreign labels, including the U.S. standard label, are
    not acceptable; stick-on labels that meet local requirements are allowed for
    products that contain a foreign label.

Prompt:

f"""Read the following text and create 5 to 10 question-answer pairs related to
    international trade law for {country_name}. Each question must include the name
     of the country. Answers should be exact quotes from the text with citations in
     the format (paragraph number, sentence number). Avoid non-trade related
    questions and duplicates.

Examples:

Question: What registration process must Brazilian importers follow according to
    Brazilian international trade law?

Answer: "Brazilian importers must register with the Foreign Trade Secretariat (SECEX
    ), a branch of the Ministry of Development, Industry, Trade and Services (MDIC)
     via its Integrated System for Foreign Trade (Siscomex)." (Paragraph 1,
    Sentence 1)

Question: What determines if additional documentation is required for imported
    products in Brazil?

Answer: "Depending on the product, Brazilian authorities may require additional
    documentation." (Paragraph 1, Sentence 2)

Question: Which ministry controls products that may affect the human body in Brazil?

Answer: "For instance, the Ministry of Health controls all products that may affect
    the human body, including pharmaceuticals, vitamins, cosmetics and medical
    equipment/devices." (Paragraph 1, Sentence 3)

Text: {text_extract}"""
```

Table 4: Correlation of Metrics with Economic Indicators, with Statistical Significance

| Info Type | Metric | Ease of Doing Business | GDP per capita | Trade % of GDP |
|---|---|---|---|---|
| -customs-regulations | is_correct | -0.277 (0.032) | 0.143 (0.322) | 0.210 (0.151) |
| -customs-regulations | completeness_bool | -0.412 (0.001) | 0.266 (0.062) | 0.287 (0.048) |
| -customs-regulations | specificity_bool | 0.023 (0.863) | 0.235 (0.100) | -0.157 (0.285) |
| -import-requirements-and-documentation | is_correct | -0.348 (0.002) | 0.315 (0.011) | 0.149 (0.277) |
| -import-requirements-and-documentation | completeness_bool | -0.306 (0.008) | 0.419 (0.001) | 0.120 (0.382) |
| -import-requirements-and-documentation | specificity_bool | 0.029 (0.806) | -0.051 (0.689) | 0.072 (0.603) |
| -import-tariffs | is_correct | -0.427 (0.000) | 0.241 (0.036) | 0.139 (0.243) |
| -import-tariffs | completeness_bool | -0.434 (0.000) | 0.268 (0.019) | 0.176 (0.138) |
| -import-tariffs | specificity_bool | 0.186 (0.081) | -0.072 (0.538) | -0.157 (0.285) |
| -prohibited-restricted-imports | is_correct | -0.286 (0.235) | 0.113 (0.701) | 0.378 (0.183) |
| -prohibited-restricted-imports | completeness_bool | -0.283 (0.241) | 0.080 (0.785) | 0.478 (0.084) |
| -prohibited-restricted-imports | specificity_bool | -0.063 (0.799) | 0.413 (0.142) | -0.602 (0.023) |
| -standards-trade | is_correct | -0.324 (0.002) | 0.139 (0.233) | 0.044 (0.709) |
| -standards-trade | completeness_bool | -0.321 (0.002) | 0.145 (0.210) | -0.037 (0.753) |
| -standards-trade | specificity_bool | 0.181 (0.093) | -0.054 (0.644) | -0.084 (0.480) |
| -temporary-entry | is_correct | -0.368 (0.003) | 0.198 (0.151) | 0.035 (0.803) |
| -temporary-entry | completeness_bool | -0.384 (0.002) | 0.267 (0.051) | 0.133 (0.347) |
| -temporary-entry | specificity_bool | 0.288 (0.022) | -0.200 (0.148) | -0.011 (0.941) |
| -trade-agreements | is_correct | -0.176 (0.121) | 0.105 (0.396) | -0.078 (0.569) |
| -trade-agreements | completeness_bool | -0.203 (0.073) | 0.185 (0.134) | -0.020 (0.883) |
| -trade-agreements | specificity_bool | -0.004 (0.974) | -0.111 (0.370) | 0.027 (0.846) |
| -trade-barriers | is_correct | -0.455 (0.000) | 0.282 (0.019) | 0.267 (0.033) |
| -trade-barriers | completeness_bool | -0.292 (0.009) | 0.283 (0.019) | 0.109 (0.393) |
| -trade-barriers | specificity_bool | 0.245 (0.029) | -0.220 (0.069) | -0.224 (0.076) |

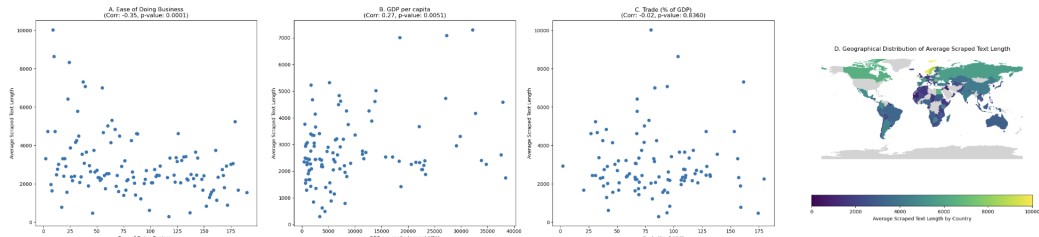

Figure 4: TradeGov Dataset Evaluation: Avg. Country Scraped Text Length

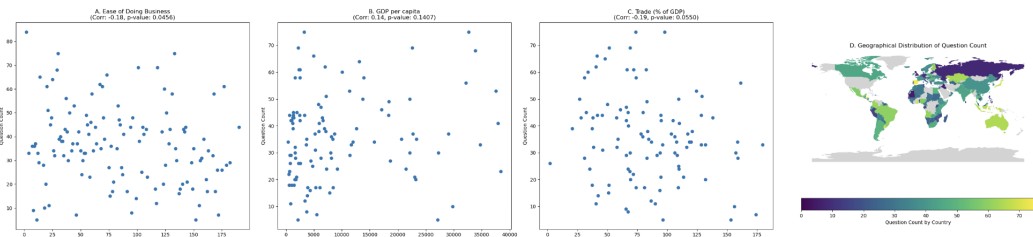

Figure 5: TradeGov Dataset Evaluation: Country Question Count

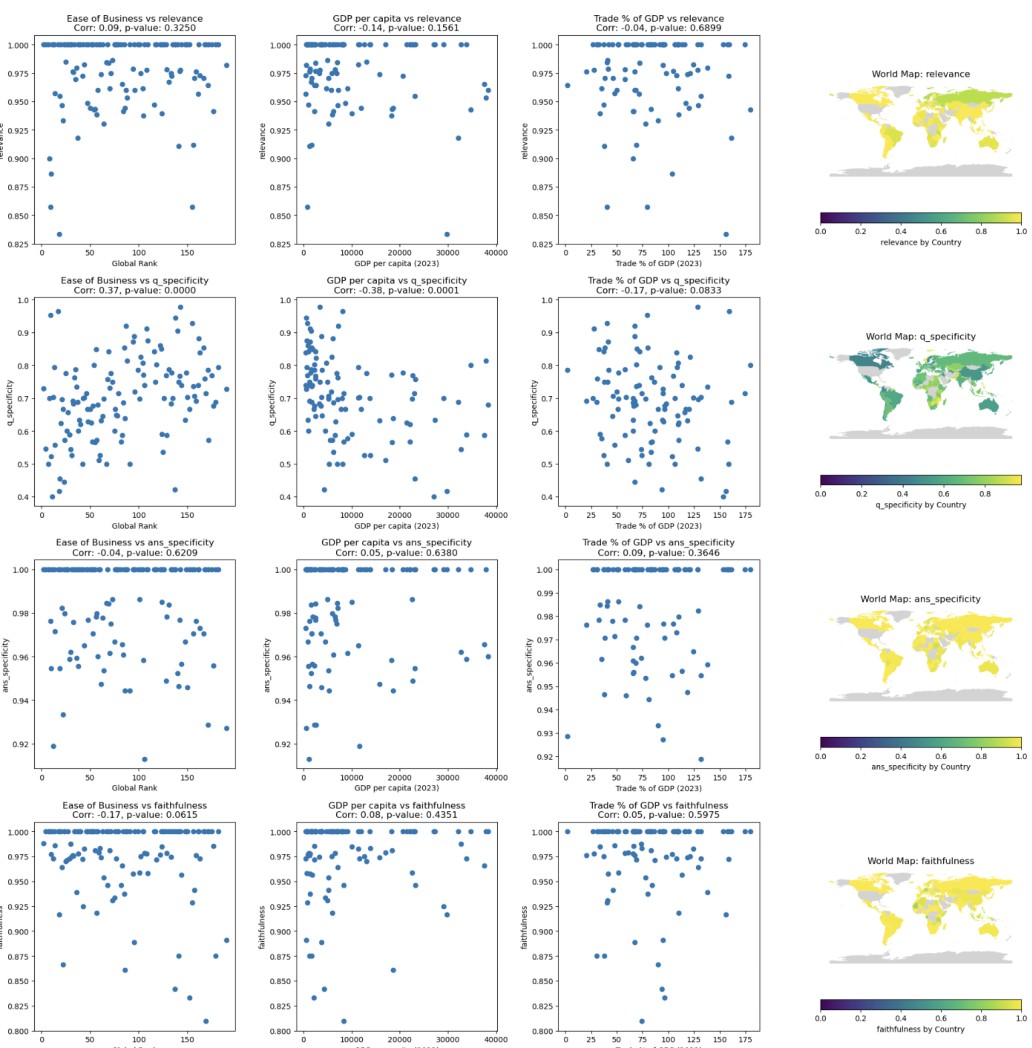

Figure 6: TradeGov Dataset Evaluation: Macroeconomic Analysis

Table 5: TradeGov Dataset : Sample Q&As

| Questions | Answers |
|---|---|
| With which agency must products that affect the human body be registered in Brazil? | Such products must be registered with Brazil's Health Regulatory Agency, ANVISA. (Paragraph 1, Sentence 5) |
| What is the VAT rate on all imports and domestically manufactured goods in Korea? | Korea has a flat 10 percent Value Added Tax (VAT) on all imports and domestically manufactured goods. (Paragraph 3, Sentence 1) |
| What is the purpose of the CE Mark in Turkey's international trade law? | The CE Mark was established by the EU to ensure products circulating within Europe met certain health, safety, consumer, and environmental protection standards. (Paragraph 2, Sentence 2) |
| Which organizations certify the quality of most non-medical goods in Zimbabwe? | The Standards Association of Zimbabwe and Bureau Veritas certify the quality of most non-medical goods produced or imported into the country. (Paragraph 1, Sentence 3) |

*(Continued on next page)*

| Questions | Answers |
| --- | --- |
| Which ministry in Vietnam publishes a list of goods with HS codes in the Import and Export Tariffs? | The Ministry of National Defense publishes a list of goods with HS codes in the Import and Export Tariffs. (Paragraph 2, Sentence 1) |
| Are Certificates of Origin required for U.S. goods imported into Ireland? | No, Certificates of Origin are not required for U.S. goods. (Paragraph 4, Sentence 10) |
| What is the role of the Uzbek Agency for Technical Regulation in Uzbekistan? | The Uzbek Agency for Technical Regulation is responsible for certification and standardization policy. (Paragraph 3, Sentence 1) |
| How long is an import license valid for motor vehicles in Uruguay? | An import license is valid for 60 days (90 days for motor vehicles) after approval. (Paragraph 1, Sentence 8) |
| How is VAT charged on imported goods in the UK? | VAT is charged as though it is a customs duty. (Paragraph 2, Sentence 3) |
| What document details the commodity codes for VAT in the UK? | VAT liability is ascertained using 'commodity codes,' detailed in the 'UK Trade Tariff: Volume 1' from HMRC. (Paragraph 3, Sentence 1) |
| What are the three rates of import duties in Ukraine's tariff schedule? | Ukraine's import tariff schedule includes Full, Most Favored Nation (MFN), and Preferential rates. (Paragraph 2, Sentence 1) |
| What does Brazil's conformity assessment system follow? | Brazil's conformity assessment system follows ISO guidelines. (Paragraph 3, Sentence 2) |
| How does Tunisia calculate VAT on imported goods? | VAT is calculated on the base price plus import duties, surcharges, and consumption taxes. (Paragraph 1, Sentence 12) |
| What system does Thailand use for import classification? | Thailand classifies imports using the Harmonized System (HS). (Paragraph 2, Sentence 2) |
| How many Free Trade Zone (FTZ) authorities exist in Singapore? | Singapore has three FTZ authorities: PSA Corporation Ltd, Jurong Port Pte Ltd, and Changi Airport Group. (Paragraph 3, Sentence 1) |
| Are tariffs on U.S. imports the same as those on EU imports in Serbia? | No, tariffs/duties on U.S. imports differ from those on EU imports. (Paragraph 2, Sentence 6) |
| What labeling regulations apply to food in Serbia? | The Rulebook on Declaration, Labeling, and Advertising of Food (RS OG No. 19/17 and 16/18) defines food labeling regulations. (Paragraph 3, Sentence 1) |
| How can low-value commercial samples be imported into Poland? | Zero or low-value samples can be imported duty-free with a written statement confirming their value. (Paragraph 1, Sentence 4) |
| What documents are needed for customs clearance in Nigeria? | Required documents include a bill of lading, commercial invoice, exit note, Form 'M' entry declaration, packing list, single goods declaration, and a product certificate. (Paragraph 3, Sentence 1) |
| When were import quotas on yellow corn and pork phased out in Nicaragua? | Import quotas on yellow corn and pork meat were phased out in 2020. (Paragraph 1, Sentence 10) |
| Where can a list of prohibited items and HS codes for Mexico be found? | The list is available on the Prohibited Items List at the Mexican Customs website. (Paragraph 1, Sentence 9) |
| What does the Mauritius-Turkey free trade agreement cover? | The agreement allows duty-free access for industrial products and specific agricultural products, including chilled fish and tropical fruits. (Paragraph 1, Sentence 16) |
| What duty is assessed on tobacco products in Kuwait? | Tobacco products are subject to a 100% duty. (Paragraph 2, Sentence 5) |
| At what stage is labeling not required for imports in Japan? | Labeling is not required at customs clearance but at the point of sale. (Paragraph 1, Sentence 2) |

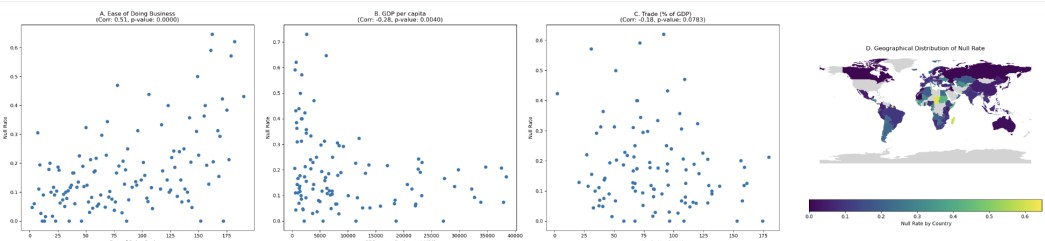

Figure 7: Null Rate Analysis - ChatGPT 4o

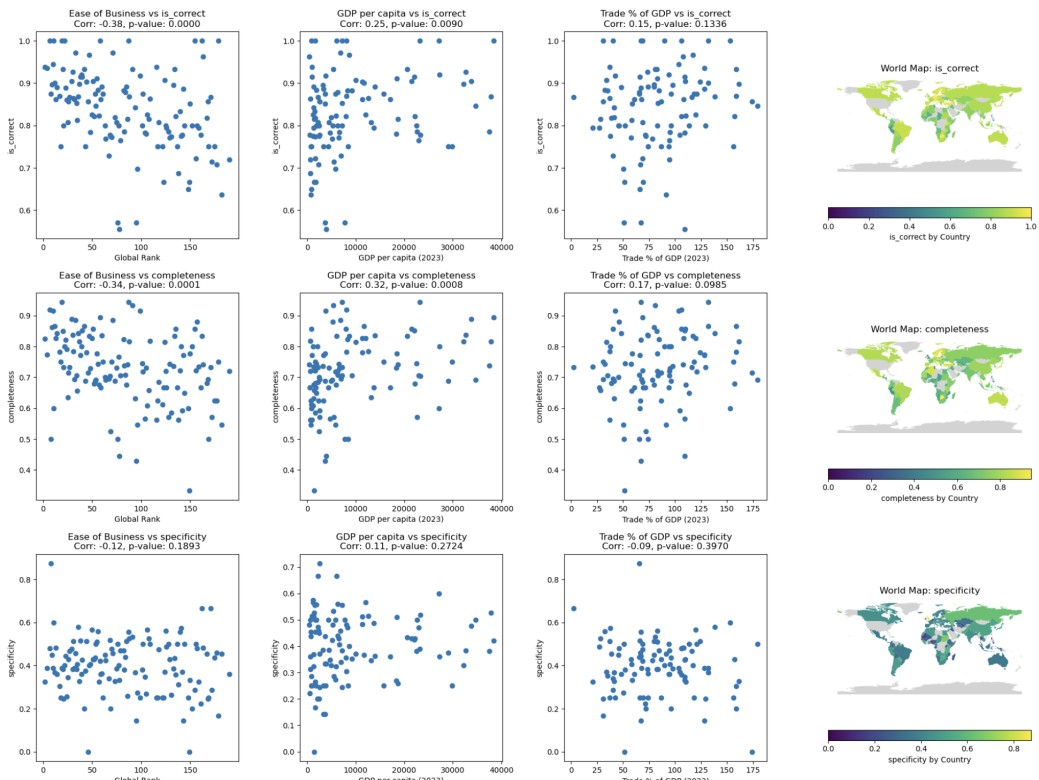

Figure 8: Response Quality Analysis - ChatGPT 4o

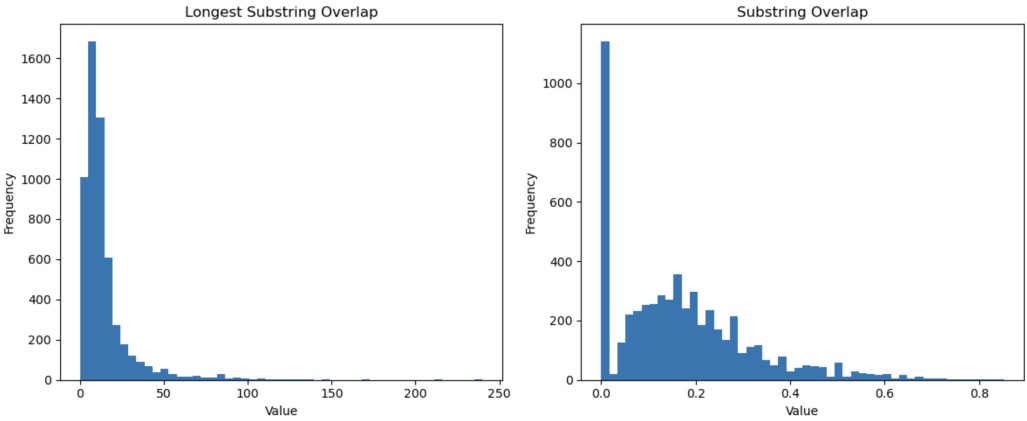

Figure 9: Memorization Quantification - ChatGPT 4o

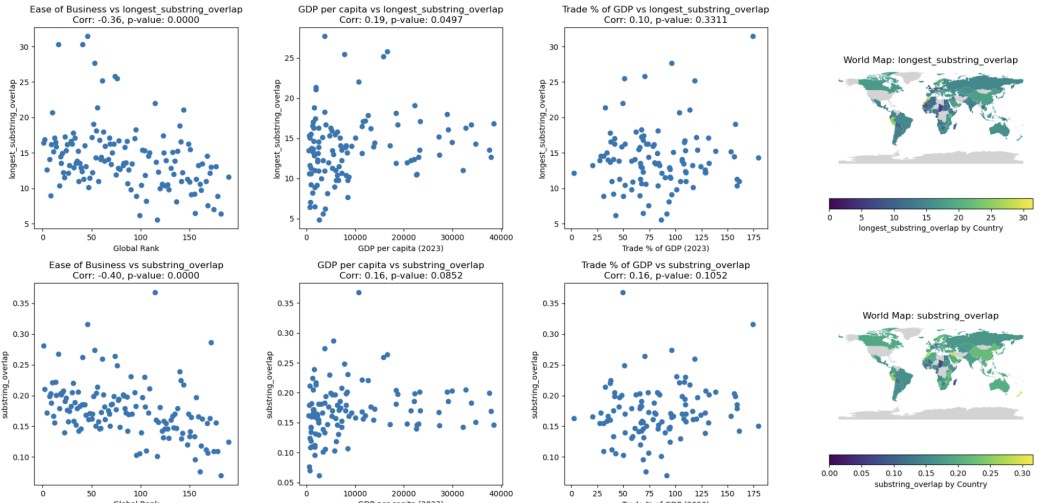

Figure 10: Memorization Quantification Analysis - ChatGPT 4o

