# OpenReview forum: "What Do LLMs Understand About International Trade? Introducing TradeGov Dataset for International Trade Q&A Evaluation"
_EurIPS.cc/2025/Workshop/UPLB — UPLB2025_

### Official Review · Reviewer_DV13 · 2025-10-27
**Useful dataset on international trade but limited overlap with the workshop**

**Rating:** 5
**Confidence:** 2

**Review:**

The paper is interesting as it introduces a new dataset to test LLMs on international trade. The authors nicely describe the dataset and test it for biases. They also show that GPT 4 displays biases when tested on this dataset. Although the paper is well written and appears to be technically correct, I'm not sure it aligns with the goals of the workshop. The paper main points are not about biases and the discussion is driven by heuristic methods.

---

### Decision · Program_Chairs · 2025-11-03

Accept (Poster)